# Decoding LncRNAs

**DOI:** 10.3390/cancers13112643

**Published:** 2021-05-27

**Authors:** Lidia Borkiewicz, Joanna Kalafut, Karolina Dudziak, Alicja Przybyszewska-Podstawka, Ilona Telejko

**Affiliations:** Department of Biochemistry and Molecular Biology, Medical University of Lublin, 20-059 Lublin, Poland; joannakalafut@umlub.pl (J.K.); karolinadudziak@umlub.pl (K.D.); alicjaprzybyszewska@umlub.pl (A.P.-P.); ilonatelejko@umlub.pl (I.T.)

**Keywords:** lncRNA, gene regulation, miRNA sponge, biomarkers, gene therapy

## Abstract

**Simple Summary:**

Advances in RNA sequencing (RNA-seq) have led to the identification of long non-coding RNAs (lncRNAs). Molecular studies on these molecules have shown that lncRNAs act as important regulators of gene expression at the transcriptional and post-transcriptional level, in both physiological and pathological conditions, yet cell functions of many of identified lncRNAs remain unknown. Here we summarize the achievements on lncRNAs studies including identification of lncRNA interactomes, structural studies and creating reporters for lncRNA activity. We also collect recent data on the involvement of lncRNAs in diseases, the clinical applications of these molecules and discuss major problems remaining the area of lncRNAs pointing future challenges.

**Abstract:**

Non-coding RNAs (ncRNAs) have been considered as unimportant additions to the transcriptome. Yet, in light of numerous studies, it has become clear that ncRNAs play important roles in development, health and disease. Long-ignored, long non-coding RNAs (lncRNAs), ncRNAs made of more than 200 nucleotides have gained attention due to their involvement as drivers or suppressors of a myriad of tumours. The detailed understanding of some of their functions, structures and interactomes has been the result of interdisciplinary efforts, as in many cases, new methods need to be created or adapted to characterise these molecules. Unlike most reviews on lncRNAs, we summarize the achievements on lncRNA studies by taking into consideration the approaches for identification of lncRNA functions, interactomes, and structural arrangements. We also provide information about the recent data on the involvement of lncRNAs in diseases and present applications of these molecules, especially in medicine.

## 1. Introduction

Non-coding RNAs (ncRNAs) were considered as superfluous by-products due to the lack of direct involvement in translation. It is now clear that these molecules play important roles in fine-tuning cellular functions. NcRNA are generally classified into two groups: those longer than 200 nucleotides, long non-coding RNAs (lncRNAs), and those below—small non-coding RNAs (sncRNA) [1,2].

Despite their extraordinary numbers, there are more than 50,000 annotated lncRNA genes in the human genome [3], lncRNAs were considered transcriptional noise until rather recently. LncRNAs are expressed at very low levels and show more cell type- or tissue-specific expression patterns than mRNAs. The biogenesis of lncRNAs is similar to that of mRNAs, where transcription, splicing and polyadenylation are mediated through RNA polymerase II [4]. The heterogenicity of lncRNAs is further enriched by the existence of isoforms through post-transcriptional alternative cleavage, alternative (or absence of) polyadenylation and/or alternative splicing [5,6,7,8]. Based on their genomic localisation, lncRNAs can be classified into intronic (transcribed from an intron within a protein-coding gene), intergenic (lincRNA; between two protein-coding genes) or enhancer (eRNA; transcribed from genomic regions distant to gene transcription start site that positively regulate nearest genes’ expression), in addition of being sense, antisense or bidirectional in reference to neighbouring genes [5,9].

In contrast to the small non-coding RNAs (sncRNAs), lncRNAs are poorly evolutionarily conserved (sequence-wise) and their cell functions are highly heterogenous [9]. Some lncRNAs affect chromatin structure by dexterously interacting with both DNA and chromatin-modifying proteins creating scaffolds for DNA-protein complexes [9], other lncRNAs can bind neighbouring genomic loci from their place of transcription to initiate genomic imprinting. Large lincRNA can also control gene expression by recruiting enzymes participating in histone modifications [10]. Additionally, lncRNAs can regulate translation, splicing and RNA stability through interaction with mRNAs [9]. Some lncRNAs seem to work as sponges inhibiting the activity of sncRNAs i.e., microRNAs (miRNAs) [11,12], and some bind mRNA or proteins [13,14,15] what results in becoming stabilisers/degrons, translocators or modulators of their activity [16,17,18]. The interaction of lncRNAs with all other macromolecules is achieved through structural recognition and/or base-pairing [5], making lncRNA either decoys, signals or guides [9]. Of note, some lncRNAs have also been found to encode peptides within small ORFs (smORFs; containing less than 100 codons) [19,20].

Furthermore, only around 1% of human known lncRNAs have been characterised to date. Progress in this field is difficult due to their limited expression in the cell, low level of lncRNA sequence conservation, and a large variety of mechanisms of action [21].

Furthermore, processes like regular co-transcriptional splicing [22] or post-transcriptional back-splicing [23,24] can produce another class of lncRNAs—the circular RNAs (circRNAs). The back-splicing circRNA can be formed from within an intron (ciRNA, circular intronic RNA), one or more exons and exon fragments with intron (elciRNA) [25]. The differences in biogenesis of circRNA might be important for their localization and thus functions, for example ciRNAs as well as elciRNAs mainly accumulate in the nucleus and are thought to regulate transcription [26,27], while exonic circRNAs are mostly present in the cytoplasm where they seem to act in post-transcriptional gene regulation e.g., as miRNA-sponges [25,28].

LncRNAs can also be classified by the function they perform—imprinted lncRNAs, disease-associated lncRNAs, pathogen-induced lncRNAs, miRNA sponges and bifunctional RNAs [10]. Imprinted lncRNAs have an important role in reinforcing local chromatin organisation, resulting in one of the autosomal alleles of a gene being epigenetically silenced [29]. Disease-associated lncRNAs are those whose expression is postnatally silenced in most tissues but re-activated during regeneration or pathophysiological conditions such as tumorigenesis [30]. Pathogen-induced lncRNAs are modulated as a response to invading microorganisms, such as *Helicobacter pylori*, and *Salmonella enterica* [31,32]. Bifunctional lncRNAs can have more than one role in gene expression and, in some cases, have smORFs [33].

This review outlines the available methods and tools currently used to study lncRNAs biology in terms of structure, interactome, activity and function. We have also summarized the mounting data on the potential applications of lncRNAs especially in medicine.

## 2. LncRNAs Structure-Functions

It is assumed that ncRNAs functions are “encoded” within their structure. In lncRNAs, numbers of local and long-range pairing of nucleotides lead to folding the strand into dynamic and flexible shapes. In this way, different structural motifs like helices, terminal or internal loops, junctions, pseudoknots, triplexes or G-quadruplexes are formed. Depending on such structural elements, RNAs are involved in *cis*- (within the same RNA molecule) or *trans*-interactions (with other molecules: RNAs, DNAs, proteins, lipids) [15,34]. Most structural analyses are based on in silico or biophysical techniques (like X-ray crystallography, NMR spectroscopy or more recently, cryo-electron microscopy). However, 3D analyses have a number of limitations for studying large molecules. First, as they require isolated and purified molecules, their assessments are based on thermodynamically stable structures, and finally the dynamic shapeshifting nature of lncRNAs is lost. In fact, in vitro or in silico studies show more lncRNA structural motifs for the same lncRNA molecule than those observed in cells [35]. This is due to the lncRNAs’ interactions with cellular factors (RNA binding proteins -RBPs, RNA helicases, ribosomes) that promote RNA conformational changes [35], often resulting in the formation of multicomponent complexes [36]. In an elegant study, Uroda and collaborators mapped the tertiary structure motifs of *MEG3* (human maternally expressed gene 3), a tumor suppressor lncRNA that modulates the p53 response, and associate such motifs with function. The pseudoknots, or “kissing loops”, of *MEG3* that are formed by the interaction between two distal complementary motifs, are critical to the interaction and stimulation of the p53 signalling pathway [37].

In addition to cellular context and environmental conditions such as temperature, ATP depletion, metabolite concentration and availability of partnering molecules, all known to act on the structure of RNAs [38,39,40], lncRNAs undergo a series of post-transcriptional modifications that also affect their affinity for binding partners. For example, unmethylated and methylated *MALAT1* exhibits different conformations in a hairpin stem [41] whereas methylation of *Xist* reduces its interactions with protein partners [42].

Yet, predicting the structure-function of any RNA—this applies to lncRNAs as well—purely based on their nucleotide sequence has proven challenging as some of these molecules achieve incredible lengths and possess high plasticity. LncRNAs are diverse class of RNA including molecules that do not exceed the arbitrary threshold of 200 nt and ‘macroRNAs’ extending beyond 90 kb. The first are *BC1* and *snaR,* which are less than or close to 200 nt but function either as primary or spliced transcripts, independent of extant known classes of small ncRNAs [43], while the second refers to 108 kb *Air*, the 91 kb *kcnq1ot1* and enormous *XACT* (252 kb) [29,44,45]. These diversity is also reflected with the stucture and function of lncRNAs. Similarly to sncRNAs, some lncRNAs act via their primary structure, for example *½ sbsRNA* containing *Alu* element necessary for binding with mRNA in STAU1-mediated mRNA decay [46] or lncRNAs sequestering miRNAs (see below).

On the other hand for some RNAs the secondary structure might be more important than the sequence. This is reflected by analyses of internal ribosome entry sites (IRES), RNA sequences responsible for cap-independent translation initiation, that show little sequence homology but similar secondary structures, supporting the idea that their plasticity and the secondary structure stability are more important than sequence [47,48]. Further, mutational analyses identified base pairs critical for IRES function, but also found that compensatory mutations that regained secondary structures could partially rescue translational efficiency [49]. Although, to make things even more complex, some IRES have no predictable secondary structure and despite that they remain functional [50,51].

The first experimentally derived secondary structure of an intact lncRNA was reported by Novikova and collaborators for human steroid receptor RNA activator (*SRA*) [52]. This 0.87 kb lncRNA posses an intricate and highly structured two-dimensional architecture organized into four major subdomains, 25 helices, 16 terminal loops, 15 internal loops and 5 junction regions. Generally, the longer the RNA sequence, the more alternative folding choices are present what makes structural predictions more challenging. For example to analyse partial of entire structure of *XIST* which is of 17 kb, an effort of many groups was undertaken [53,54,55,56] (see below). Recently, the modular domain architecture of *XIST* in complex with interacting proteins was revealed [57]. Interestingly, discrete *XIST* domains are responsible for binding of distinct sets of effector proteins nonetheless the central role of the A-repeat domain in this process is indisputable.

While the obvious and simplest manner of evaluation of lncRNAs by sequence similarity comparison didn’t reveal features that survived through generations, comparing lncRNAs’ genome locations, structure (exon-intron), the 3D structure of the resulting RNA, and their expression patterns, support that probably over 70% of 5413 human lncRNA analysed are evolutionary conserved [58]. The comparison of the structure-function of these many lncRNAs is an arduous task, yet, in one seminal study, Hezroni and collaborators analysed the most well-known human lncRNAs and concluded that more than 1000 lncRNAs have similar functions in other mammals [59]. Based on these facts, we suggest that lncRNAs should be analysed with view of structure shifting in a landscape of biological factors, similar to the way intrinsically disordered proteins are investigated [60,61].

For the above-mentioned reasons, new or adapted strategies to map the motifs and functions of lncRNAs had to be developed. Today, the general rule in lncRNA structure analysis is based on six stages: (1) in vivo/in vitro experimentation; (2) RNA denaturation/purification; (3) probing—enzymatic/chemical; (4) sequencing (High-Throughput Sequencing -HTS/Next Generation Sequencing -NGS), RNA-seq/capillary electrophoresis; (5) bioinformatic analysis; (6) determination of nucleotide position and modification frequency.

## 3. Structural Analyses of lncRNAs

The preferred tool to study the structure of lncRNAs is enzymatic/chemical probing, supported by computational RNA structure prediction methods (Figure 1) [62]. Although, other methods exist (see Table 1). The use of chemical probes or nucleases is based on the fact that RNA bases, whether they are free, pairing, or forming 3D structures, show different accessibility and reactivity to different types of probes. Binding of small molecule probes to RNAs leads to formation of adducts that cause reverse transcriptase (RT) to either create errors recognized as ’mutations’ or ’stop’ events during reverse transcription. The first results in modifications of newly synthesized cDNA while the latter causes the detachment of RT from the RNA fragment and a truncated cDNA [63,64]. These modifications are then positioned using capillary electrophoresis or sequencing techniques where cDNAs are compared with probed and untreated RNAs.

Enzymatic probing utilises nucleases recognising and cleaving specific sequences or structures e.g., single- or double-stranded regions (RNases such as P1 and V1, respectively), A’s (RNase A) and U’s (RNase U2) [63], which were later subjected to reverse transcription and sequencing. Both of these methods can be applied to purified lncRNA but only chemical probing is used in living cells (in vivo) due to the difficulties with the identification of lncRNAs secondary structures at the single-nucleotide level after nuclease digestion of millions of RNAs present in the cell. In general, data obtained after digestion with one nuclease shows only a structural trend (e.g., single- or double-stranded character), but in most cases, additional analysis using other nucleases or chemical probes are necessary [63].

In contrast, small molecule probes easily penetrate into living cells and provide better resolution [63,65]. In 2013, Novicova and collaborators described a new approach called 3S (Shotgun Secondary Structure) that is especially useful for analysing long RNAs [66]. It is based on parallel chemical probing of the entire lncRNA as well as multiple fragments of the lncRNA overlapping with each other. LncRNA fragments are prepared by in vitro transcription from dsDNA templates obtained from available cDNA clones or chemically synthesized by custom gene services. The obtained probing profiles of fragments are compared with the profile of the full RNA to estimate the identity/similarities. Regions with profiles similar to the full-length transcript are considered to have the same secondary structure. As the base-pairing partners within this region are not likely to occur outside the region, this method enables to map the structure of RNA through hierarchical probing of smaller and smaller fragments of the full RNA. 3S has been used for determination of the secondary structures of the best known lncRNAs: *SRA*, *HOTAIR*, *COOLAIR*, *RepA*, *Braveheart* and *NEAT1* [66,67,68,69,70,71].

HTS is nevertheless considered the most specific technique to analyse lncRNA structures to date. The development of methodologies coupling HTS with ribonuclease cleavage or chemical probing facilitate RNA structure mapping in the context of whole transcriptome. For example, FragSeq (Fragmentation Sequencing) is based on P1 nuclease (which specifically degrades single-stranded nucleic acids) probing followed by HTS and bioinformatic analysis. The degradation in a site between two adjacent bases is characterized by a ‘cutting score’ reflecting the preference of P1 nuclease to digest comparing to other sites in the same RNA; as P1 cuts 3′ of an unpaired base, the uncleaved nucleotide targets may be engaged in base pairing or tertiary interactions [72]. The ds-RNA fragments left after nuclease-treatment are further separated according to size, cloned together with 5′ and 3′ specific adapters and sequenced. The obtained reads are used in mapping of single-stranded of RNA regions by comparing the sequence with sequencing libraries of multiple ncRNAs with known structure.

## 4. Identification of lncRNAs’ Binding Partners

Recently, initiatives such as lncRNA interactome mapping, single-complex mapping, and large-scale (network) mapping have been used for revealing the complexity of lncRNAs partnerships with other RNAs, proteins and/or DNA and thus provided some information about their potential functions.

### 4.1. RNA-RNA Interactions

Interactions between pairing RNAs, or duplexes within the same molecule, are most commonly detected following one of two strategies: crosslinking of the two RNA strands or employing known RBPs. The former approach is used in techniques such as PARIS (Psoralen Analysis of RNA Interactions and Structures) [78], SPLASH (Sequencing of Psoralen crosslinked, Ligated, and Selected Hybrids) [79], and LIGR-seq (LIGation of interacting RNA followed by high-throughput sequencing) [80], where a photochemical reagent—psoralen, or its derivates, is used to (Figure 2A) intercalate and crosslink double stranded fragments of RNA (or DNA) molecules. Photoirradiation results in the formation of covalent adducts between psoralen and pyrimidine bases. An advantage of psoralen is that it is activated at higher wavelengths than standard UV light crosslinking (320–400 nm for psoralen vs. 260 nm for UV), causing less damage to nucleic acids.

Furthermore, the bridge formed between two RNA fragments is photoreversible by short wavelength (264 nm) UV light [81,82]. Psoralen crosslinking is followed by purification of RNA duplexes, ligation of adjacent nucleic acids termini (proximity ligation) and identification of interacting RNAs in *cis* and in *trans*, by sequencing. For instance, PARIS was used for analysis of structural organization of *XIST*, a nuclear lncRNA that triggers silencing of one of the X chromosomes in mammalian female cells, showing that four major coherent domains are kept by local duplexes within *XIST* [78]. Additionally, the authors reported the interactions between snRNA *U1* and lncRNA *MALAT1*. The main drawback of psoralen-dependent methods is the relatively low number of uniquely mapped duplexes due to psoralen’s limited efficiency [35].

The second strategy includes pull-down of tagged RBP in complex with their interacting RNA(s). Most commonly used RBPs are Argonaute proteins [83], PUM2 (human Pumilio 2, a member of the Puf-protein family), QKI (Quaking homolog, START proteins family) [84], and small nucleolar RBPs [85] (see also Table 2), that bind sequence or structure-specific regions of RNA. The RBP-mediated methods (Figure 2B) such as CLASH (Cross-linking, Ligation And Sequencing of Hybrids) [86] and hiCLIP (RNA hybrid and individual-nucleotide resolution ultraviolet crosslinking and immunoprecipitation) [87], follow a similar strategy starting by overexpressing a known RBP that binds to RNAs duplexes. These methods have assisted the discovery of new snoRNA-rRNA interactions in yeast, and miRNA–mRNA dimers recognised by Argonaute in human cells [88], respectively. It is unknown whether overexpression of tagged-RBPs, although innovative, influence RNA–RNA–RBP interactions. MARIO (MApping RNA Interactome In vivo) [89] overcame such potential problem by UV light-crosslinking of any RNA–protein complexes, followed by RBP labelling by covalent biotin binding to amino acids such as lysine or tyrosine, or to carboxyl group on the C-terminal ends of proteins using biotin ligases. The RNA is then fragmented by RNase I, and the RNA–RBPs are then immobilised on streptavidin/avidin beads. The adjacent RNA ends are further joined by short biotin tagged RNA linkers. Next, biotin labelled complexes are purified, and RNAs are extracted for sequencing. MARIO has been used to capture the RNA interactome, including lincRNAs, based on the whole proteome [89].

### 4.2. RNA-DNA Interactions

The first eukaryotic lncRNA (*H19*) was identified in the pre-genomic era during cDNA library screens performed to study gene expression. Initially, it was considered an mRNA due to the presence of small ORFs [93]. Despite the fact that *H19* does not code any protein, its dosage is important in embryonic development [94]. Shortly afterward, another lncRNA participating in dosage compensation, *Xist*, was identified as X-inactive-specific transcript, during the pioneering studies on X-chromosome inactivation (XCI) in mice [95]. Further studies on *H19*, *Xist* and newly identified lncRNAs revealed tight interplay between lncRNAs and DNA.

Evidence suggested that one of the most prominent roles of, at least some, lncRNAs was the regulation of chromatin structure [96], and several techniques were succesfully applied to decipher the molecular mechanisms of lncRNAs action. For example, Chromatin Isolation by RNA Purification (ChIRP), using hybridising biotinylated oligonucleotides, showed that *HOTAIR* acts as an active recruiter of chromatin-modifying complexes and is associated with relocalisation of PRC2 (Polycomb repressive complex 2)—a protein complex with H3K27 histone methytransferase activity that causes chromatin condensation and gene silencing. The lncRNA-dependent chromatin occupancy of PRC2 has been further confirmed by Long and collaborators [97], who reutilised rChIP (RNA-dependent Chromatin Immunoprecipitation) and coupled it with deep sequencing (rChIP-Seq).

Modifications of these techniques exist to improve specificity, for example by an initial RNase-H mapping assay to determine the most accessible sequences for probing. To map the chromatin interaction sites of lncRNAs, a ChIRP like, but more specific, method is commonly used: capture hybridization analysis of RNA targets (CHART), which uses short (22–28 nts) antisense biotinylated-oligonucleotides as probes to capture lncRNA-chromatin complexes [98]. This approach revealed that *roX2*, a *Drosophila* lncRNA, binds across gene bodies of X-linked genes exhibiting focal peaks of high occupancy at chromosomal entry sites [99]. In addition, RNA Antisense Purification (RAP) [53] uses antisense RNA probes (a bit longer than ChIRP: 120 nt for *Xist*), which improves specificity and minimizes background noise. RAP analyses showed that during maintenance of XCI, *Xist* is localized broadly across the entire X chromosome, lacking focal binding sites [53].

A significant limitation of all the above-mentioned methods is the need of knowing the sequence of the targeted nuclear lncRNA (nlncRNA). To overcome this obstacle, three new methods have been developed, namely: MARGI (MApping RNA-Genome Interactions) [100], GRID-seq (Global RNA Interactions with DNA by deep Sequencing) [101], and ChAR-seq (Chromatin-Associated RNA Sequencing) [102]. The common feature between them is the crosslinking of nlncRNAs with their DNA targets, followed by proximal ligation, fragmentation, and sequencing. The outcome is a detailed and comprehensive database of genome-wide interactions, showing that most of the nlncRNAs act as *cis*-regulators, and only a few notable nlncrNAs such as *MALAT1*, *NEAT1*, and *roX2* are *trans*-regulators [101].

### 4.3. RNA-Protein Interactions

RNA–protein interactions can be studied either through the prism of RNA or that of protein. RNA-centric methods employ probes in similar ways as in RNA-RNA analyses [95], yet they differ in the downstream steps, usually involving mass spectrometry (MS) to determine protein identities. One such method is ChIRP-MS (Comprehensive Identification of RNA-binding Proteins by MS) [103], which captures lncRNA(s) by biotinylated oligonucleotides together with their binding proteins, and are identified by MS. The technique was successfully used to identify 81 *Xist*-interacting proteins, among them HnrnpU and HnrnpK emerged as the most abundant *Xist*-associated factors, and both functionally contribute to XCI [104]. The obvious goal of these methods is the identification of proteins interacting with a given RNA, which means to distinguish between covalently bounded RNA–protein and non-covalent interactions. In RAP-MS (RNA Antisense Purification coupled with quantitative MS) the purification of complexes is performed in denaturing and reducing conditions to disrupt the non-covalent interactions. Therefore, the mapping of specific RNA required long biotinylated probes, forming very stable RNA–DNA hybrids. Another challenge has been the quantification of interacting proteins vs. background proteins, what might be accomplished by isotopic labelling of amino acids and quantitative comparison of purified proteins by MS. At the end a short list of high confidence interactions is given, for example *Xist* interacts with SHARP and SMRT proteins, recruiting HDAC3 (Histone Deacetylase 3), and with PRC2 guiding the complex to the X chromosome for transcriptional silencing [105].

On the other hand, pulling down a known RBP with its lncRNA in complex with other proteins has been successfully carried out using two main approaches. The first involves the modification of the lncRNA, inserting an RNA sequence that is known to attract a specific RBP. The second option is to use a known RBP as a lure to pull down lncRNA–protein complexes. For example, the MS2 bacteriophage coat protein (MCP) tagged with the HB (histidine-biotin) tag, a peptide sequence that contains 6xHis and the 75-amino acid sequence of *Propionibacterium shermanii*’s transcarboxylase—which is efficiently biotinylated in eukaryotic cells by endogenous biotin ligases [106] has been used to pull down MS2 loop-tagged lncRNAs together with their interacting RBPs, a technique named MS2 in vivo Biotin Tagged RNA Affinity Purification (MS2-BioTRAP) [107]. The application of HB-tag allows to purify authentic complexes of MS2-RNA–RBPs [107]. As in all cases for protein identification, MS is the top choice.

In order to retain the RBP–lncRNA complexes, UV light cross-linking is most commonly used [108,109], followed by immunoprecipitation (CLIP) [110] and numerous derived methods such as high-throughput sequencing HITS-CLIP [111] for comprehensive identification of RBP target sequences at the transcriptome level, photoactivatable ribonucleoside enhanced PAR-CLIP where alternative ribonucleosides are used during RT at the position of cross-linking, which enables the detection of RNA–RBP interacting sites with nucleotide resolution and reduces background reads [84], or enhanced CLIP (eCLIP) where 3′ RNA linkers and further 3′ DNA linkers are ligated to increase the total number of non-PCR reads that can be obtained after HTS, often decreased due to the RT termination at the crosslink site [112]. For more information regarding the different variants of CLIP methods, we suggest the following review [17].

## 5. Determination of lncRNA Function (Reporter Systems)

Studies on the function of lncRNA have been hindered by lack of relevant tools. Thus, most lncRNA reporter systems have been created *ab initio*. As most of the known functions of lncRNAs have been discovered in nlncRNAs, such as *XIST* and *HOTAIR*, their functions have been assigned as gene or chromatin regulators. Such nlncRNAs often work as scaffolds of unsympathetic protein partners and/or guides of RNP complexes to particular loci—what results in local changes in transcription, chromatin structure histone modifications (methylation, acetylation) and/or DNA methylation [113,114]. Therefore, targeting nlncRNAs to synthetic promoter–reporter systems is currently the most common approach to determine the function of these enigmatic nucleic acids as epigenetic modulators.

Here, we shall separate the assays into two types: those to study gene repression and those to study gene activation (Figure 3). The practical difference between the two is that for the first a constitutively active promoter is used and thus repression is measured as the decrease of reporter signal, while for the latter a transcriptional complex must be delivered to a minimal promoter for transcriptional activation and reporter expression. Either of these promoters include a binding sequence, often in tandem repeats, for a DNA binding protein (DBP). The DBP is then engineered as a fusion with the RNA binding domain (RBD) of an RBP (Figure 3A), which might be an endogenous protein known to interact with the specific lncRNA or an artificial (exogenous) protein that binds to a specific RNA sequence/structure. As mentioned above, there are RBPs to natural or synthetic loops e.g., MCP-MS2 loops, and novel proteins that are able to bind RNA by guiding RNAs (see Table 2 for a list of the most commonly used RBPs).

As endogenous RBPs often bind to a plethora of RNAs, domesticated RBDs are far more specific and thus more commonly used. Synthetic RBPs usually bind to RNA-specific loops, these loops need to be artificially cloned in fusion with the lncRNA to be studied. Two of the most common loop binding proteins are λN and MCP, that bind to the BoxB and MS2 loops, respectively. By fusing a DBP, e.g., GAL4 or TtA, to either of these loop-binding proteins, they will drag (guide) the lncRNA to either a constitutive or a minimal promoter, depending on the desired effect (repression or transcriptional activation, respectively). Such promoters control the expression of a reporter gene (luciferase or fluorescent protein) (Figure 3). These systems have been used to verify the functions of lncRNAs such as *NALT*, *LUNAR1*, *LINC00152* or *HOTTIP* as transcriptional activators [90,91,115,116], and HOTAIR as a transcriptional repressor [92] (Figure 3B).

It is of note that some lncRNAs bind directly to DNA sequences forming tertiary structures (Figure 3C). In some cases, the interaction lncRNA–DNA does not seem to be sequence-specific e.g., *XIST* or *Ferre* [53,117] who bind DNA in *cis*-, nearby its production site [118,119]; while in other cases, lncRNAs have targeted sequences. For example, the lncRNA *CISAL* contains a 22 nt DNA-recognition sequence (900 nts from its 3′-terminus) to the promoter of BRCA1. Binding to the BRCA1 promoter results in a triple helix structure, while another region of *CISAL* sequesters GABPA, a BRCA1-transcription factor, away from downstream regulatory regions, all of which uncouples BRCA1 transcription [120]. This study is of particular interest as the lncRNA-interacting sites for promoter and transcription factor were elegantly mapped by truncations and mutagenesis of all three partners (DNA, lncRNA and protein). Another example is *lnc-MxA*, a lncRNA that binds to the IFN-β promoter forming a triplex with the DNA and leading to interference of gene expression—as determined by a reporter bioassay (Figure 3C) [121].

LncRNAs as miRNA ‘sponges’ were first described in plants, where an *Arabidopsis thaliana* non-protein coding gene IPS1 (Induced by Starvation 1) harbouring a *miR-399* targeting sequence was reported to sequester that miRNA away from mRNA [122]. To found more information about sponging miRNAs by lncRNAs we recommend the following article [123], here we will only focus on the development of an assay to analyse the activity of lncRNAs being ‘sponges’ of miRNAs (Figure 3D). Since miRNAs target mRNA for degradation, a miRNA sponge works by hijacking miRNAs and thus a positive regulator of mRNAs. Using a luciferase reporter system, the target region of a miRNA, often located in the 3′-untranslated region (3‘UTR) of the target’s mRNA, is cloned downstream of luciferase, thus upon miRNA expression luciferase expression is suppressed via miRNA-mRNA destabilization. Therefore, if co-expression of a lncRNA results in miRNA binding and regain expression of luciferase, it suggests that the lncRNA is a miRNA sponge. Further verification includes mutagenesis of the miRNA-binding site in the lncRNA of interest, to ensure direct effects [124].

Tools to unravel the impact of lncRNAs on chromatin condensation are still limited due to the complexity of the process. Yet, there are some groundbreaking examples such as a doxycycline (DOX)-inducible system regulating *XIST* that was integrated into multiple *loci* (different chromosomes) to investigate the effect of local *XIST* expression in the neighbouring DNA and chromatin environment. After DOX induction, *XIST* expression and localization was determined by RNA Fluorescence In-Situ Hybridization (FISH), showing that local gene silencing could be achieved by this lncRNA without the participation of other known, and thought to be essential, chromatin modifiers such as macroH2A, SMCHD1, and H3K27me3/H4K20me1. The analyses by allele-discriminating pyrosequencing assays revealed the pattern of genes silenced flanking the integration sites, while ChIP-seq showed that *XIST* –mediated silencing occurs at all sites tested, but the range of silencing to flanking endogenous human genes is variable [118].

## 6. Mapping lncRNAs’ Functional Domains

LncRNAs, similarly to proteins, can acquire 3D structures, lncRNAs are often more dynamic shapeshifters than their amino acid counterparts, as well as having intrinsically disordered regions—similar to intrinsically disordered proteins [51]. This, turns out, allows these nucleic acids to use sequence-specific-binding to other RNAs or to DNA, to form docking structures for proteins, RNAs and DNA, and become magnets and guides for multi-component complexes between all three macromolecules. The detailed biological understanding of lncRNAs needs the identification of their functional domains in terms of specific motifs, local architecture, and/or posttranslational modifications. Analysing lncRNA is far more challenging than proteins because of the unique features of these molecules. The initial approaches to reveal lncRNA functional domains were based on the RNA sequence similarity but provided only limited useful information [125]. More informative data was obtained by analysing common structural RNA elements [126,127], which, as mentioned earlier, in many cases are evolutionary conserved [3,128]. Since the prediction of functionally relevant sequences requires careful verification by in vivo studies, a variety of methods to systematically map lncRNA functional domains are under development. Below, we summarize some of the previously used and most modern techniques.

Screening for functional molecules and/or functional domains has traditionally been performed by loss-of-function (LoF) studies. LncRNAs have been knocked-down by small interfering RNA (siRNA), endoribonuclease-prepared RNA (esiRNA) or short hairpin RNAs (shRNA) sampling for biologically relevant phenotypes (Figure 4A) e.g., stemness [4,129,130,131]. However, some of the approaches induced a large number of side-off effects, off-targeting on other transcripts, or detected only specific phenotypes in particular cell lines and/or conditions [132].

Similar to siRNA, ASOs (antisense oligonucleotides) have been applied to target certain regions of lncRNAs (Figure 4B) [133]. ASOs can hybridize with complementary RNA sequences and recruit endogenous RNase H to cleave the lncRNAs leading to either knock-down lncRNA or sterically block the access of functional domains without degradation [133]. Chemical modifications on ASOs improve their stability and reduce unwanted cell responses e.g., interferon response what makes them useful also for therapy of various diseases [134]. Importantly, ASOs can also enter the nucleus and thus effectively modulate activity of both cytoplasmic and nuclear lncRNAs [135].

Yet, the direct targeting of functional regions within a lncRNA was not done until Beletskii and co-workers developed Peptide Nucleic Acid (PNA)-Interference Mapping (P-IMP) (Figure 3C) [136]. PNAs mimic nucleic acids but having a peptide backbone, composed of charge neutral and achiral N-(2-aminoethyl) glycine units, which, unlike natural nucleic acids, protects them from nucleases [137]. Originally, this technique was used for mapping the regions of the murine *Xist*, revealing that a distinct repeat in the first exon is responsible for binding to genomic DNA [136].

Similarly, Sarma and collaborators used locked nucleic acid (LNA) technology to target and block key regions of *Xist* (Figure 3D) [138]. LNAs are nucleic acid analogues containing the ribose ring “locked” by a methylene bridge between the 2′ oxygen and the 4′ carbon, which results in higher stability and an increased affinity of LNAs to base pair with complementary RNA vs. DNA. In this way several domains of *Xist* engaged in its nuclear distribution were identified [138].

The advent of genome editing tools, in particular Clustered Regularly Interspaced Short Palindromic Repeats (CRISPR)/Cas9, brought several additional possibilities to the lncRNA field. First, it allowed for the permanent genetic modification of endogenous lncRNAs genes by targeting Cas9 nuclease using guide RNAs (gRNAs) to specific genetic sequences and introducing InDels (insertion or deletions) [139]. Indeed, systematic deletions of *NEAT1* in a human haploid cell line resulted in the identification of a modular domain architecture of this lncRNA [140,141]. The authors depicted three functional *NEAT1* domains playing distinct roles during formation of RNA–protein complexes called paraspeckles. Moreover, this approach was also effective to show three subdomains within the middle domain of *NEAT1* responsible for paraspeckles assembly [140]. Since CRISPR/Cas9 is amenable for high-throughput screening, by using libraries of gRNAs, targeting unique genomic locations, single or multiple targets can be analysed downstream phenotypical screens. These screens have been very effective to identify the functional domains of proteins and recently have been adopted to non-coding genes too. For example, Tiling CRISPR, where a set of gRNAs is used to direct Cas9 to sequences covering the entire gene of interest (Figure 5), was successfully used to authenticate the region containing A-repeats as *XIST* silencing domain [54].

## 7. LncRNAs Databases and Bioinformatic Tools

The lncRNA databases and bioinformatic toolkit can greatly assist the research on lncRNAs. Some of available online tools for in silico lncRNA analysis provide rather general information, while others are more specific. The first includes ’meta-databases’ of lncRNAs such as LNCipedia, representing the most comprehensive compendia of human lncRNA [142]. It contains data from ten unique origins as of 2019, allowing embedded prognostication of coding potential, yet there is no automatic forecast of subcellular localization, correlation with disease or functional estimates. Contrary, LNCBook compiles lncRNAs from both previously established databases and experimentally verified transcripts selected by the community [143]. LNCBook tenders multi-omic data integration such as expression profiles in normal and carcinogenic tissues, function annotation and disease association, DNA methylation patterns in various gene regions, genome variability and forecasts of microRNA (miRNA) synergies using miRanda [143,144] and TargetScan algorithms [145]. Both LNCipedia and LNCBook extend embedded coding potential foresight through predictor of lncRNAs and mRNAs based on specific pattern containing *k* nucleotides called *k*-mer (PLEK) [146] and Coding Potential Alignment Tool (CPAT) [147]. As this field is evolving rapidly and numerous new transcripts are steadily being distinguished, these databases should be frequently updated to improve their potential.

The second group of tools is dedicated for analysing particular features of lncRNAs and refers to targeted databases; for example, databases of lncRNA expression in human tissues (GTEx) [148], cancers (TANRIC) [149], and plants (CANTATAdb) [150], or lncRNA localization such as LncSLdb [151] or LncATLAS [152], and the most popular predictor of subcellular localization- LncLocator [153]. In addition, essential in lncRNAs research are tools designed to separate transcripts encoding micropeptides from authentic non-coding RNAs, such as Coding Potential Prediction (CPPred) [154] as well as PhyloCSF [155] and COME [156]. Recently, a freely available mobile-friendly web server- Coding-Non-coding Identifying Tool (CNIT) [157] based on the Coding-Non-Coding Index (CNCI) [158] database, was introduced for researchers.

At present, the only available tools for structure prediction of lncRNAs are those dedicated to all RNA molecules. One of which is the RNA Mapping Database (RMDB) that facilitate access and meta-analysis of the diverse structural mapping experiments performed on ribonucleic acids [159]. To date the use of RMDB for lncRNA may be limited, however the rapid development of high-throughput techniques may contribute to increasing the functionality of this database [160]. Ultimately, RNAfold [161] or DMfold [162] can be used to predict lncRNA structure. The RNAfold predicts minimum free energy (MFE) folding for RNA secondary structure and equilibrium base-pairing probabilities, both algorithms are considering also circRNA [163]. The limitation of RNAfold with predicting pseudoknots can be solved using DMfold, which is based on deep learning [162]. Though DMfold exhibits a high accuracy in predicting lncRNA structure, it may be less accessible due to its command line utility [160].

Despite the growing demand, the availability of in silico methods and predictive tools for mapping lncRNAs functional domains is still poor. Since single bioinformatics tools are unable to truly estimate the structure: function of lncRNAs, a bioinformatic tool—Predicting LncRNA Activity through Integrative Data-driven Omics and Heuristics (PLAIDOH)—has been created to combine data from the transcriptome, subcellular localization, enhancer landscape, genome architecture, chromatin interaction, and RNA-binding (eCLIP) analyses, providing statistically defined output scores [164]. This approach also provides the interactome between individual lncRNA, coding genes, and protein pairs using enhancer, transcript *cis*-regulatory, and RBP data to enhance its predicting capacity.

In contrast, there are several databases for predicting lncRNA partnerships. Databases, such as IntaRNA 2.0 [165] or LncRRIsearch [166] are dedicated to predict lncRNA—RNA interactions, while TheLnChrom [167] and Triplexator [168] to predict lncRNA–DNA interactions. For their detailed comparison we recommend the following article [160]. In addition, many tools focus on lncRNA–protein interaction, such as NPInter [169,170], starBase [171] and several others [172]. Additionally, RNA Interactome Repository-RNAInter is more comprehensive tool collecting information from published data along with another 35 database resources [173].

Recently, to predict lncRNA function a new approach of sequence evaluation based on comparison of short sequence elements (*k*-mer) representation (SEEKR), was developed [174]. As lncRNAs with similar *k*-mer content have been shown to exhibit related functions despite their lack of linear homology, the SEEKR database provides very useful information about some determinants of lncRNAs function. In addition, lnChrom [167] and lncRNADisease [175] databases collect lncRNA-disease associations.

## 8. LncRNAs and Disease

Since multiple lncRNAs have been found playing roles as regulatory elements in gene expression, it is of little surprise that they take part in physiological and disease processes [5]. We summarize them in the section below, with the exception of their potential role as biomarkers in cancers, which we describe in a separate chapter.

LncRNAs have rather clear and unique expression patterns, which makes them good markers for health and disease conditions. Changes in expression of many lncRNAs are specific to the tissue, developmental stage and/or conditions (see Table 3).

Many reports describe the dysregulation of lncRNAs in multifactorial chronic diseases [187]. Starting with neurodegenerative diseases such as Alzheimer [188,189,190], Parkinson’s [191] and Huntington’s [177]; through autism spectrum disorders [187,192,193]; and schizophrenia [194,195], cardiovascular diseases, such as during chronic heart failure [196,197], atherosclerosis [198,199], myocardial infarction [200], and diabetic cardiomyopathy [201]. Moreover, as lncRNAs are implicated at many levels of metabolism regulation, perturbations in their expression serve as an important component for the occurrence of metabolic diseases. In a comprehensive study, Morán and collaborators identified a series of pancreatic islet lncRNAs (e.g., *HI-LNC12*, *HI-LNC25*, *HI-LNC75*, and *HI-LNC78*) dynamically regulated during β-cell differentiation and maturation, two other (*KCNQ1OT1* and *HILNC45*) that are dysregulated in type 2 diabetes, and a set of lncRNAs that map to human diabetes genetic susceptibility loci [202]. Among other lncRNAs having diagnostic value are *TUNA* associated with the intensity of Huntington’s disease [177], *MALAT1* upregulated in myocardial infarction [203], and *Mhrt* downregulated in cardiac hypertrophy [204].

In addition, data gained from the studies on infectious diseases show that lncRNAs have been associated with the regulation of both pro- and anti-inflammatory processes [205,206,207,208]. Moreover, lncRNAs seem to play critical roles in the regulation of pathways in autoimmune diseases such as rheumatoid arthritis [209,210], psoriasis [7], and Chron’s disease [211]. Expression of lncRNAs changes in response to various pathogens—likely due to deregulation of host immune responses [212,213,214]. There seems to be a footprint pattern of lncRNAs expression depending of the pathogen, as has been shown by studies with *Escherichia coli* [215], *Salmonella typhimurium* [216], *Mycobacterium tuberculosis* [217,218,219,220], *Campylobacter oncisus* [221], or *Helicobacter pylori* [222,223], also after infection by the human immunodeficiency virus (HIV) [224], ZIKA virus, Sendai virus, hepatitis C virus [205] and COVID-19 [225].

A recent report on *Toxoplasmosa gondii* infection, for the first time, suggests that protozoan infections also alter lncRNA expression [226]. The molecular mechanisms at play by these lncRNAs during infections are yet to be discovered. Virtually no information is available on lncRNA effects during protozoan invasion, which might be an interesting area of research, in particular if there are common patterns versus infectious agent or host responses—beneficial or deleterious—as they could be potential drug targets. Considering the poor sequence conservation along lncRNAs, it will be highly interesting to compare the different species responses to infections, which might not only bring new information to human medicine, but to zoonosis and veterinary medicine, as well.

## 9. Applications of lncRNAs

### 9.1. Diagnostic Biomarkers in Cancers and Targets for Therapy

It is widely known that early detection of tumorigenesis greatly increases the chances for successful treatment and survival. Numerous studies have shown that several lncRNAs are associated with the stage and prognosis of multiple tumour types [31,227,228,229,230] such as breast [231,232,233,234], lung [235,236], gastric [75], colorectal [237,238], and prostate cancers [239]. Additionally, the overexpression of multiple lncRNAs e.g., *AC006050.3-003* and *ODRUL* in cancer cells increases their resistance to DNA damaging agents [240,241], suggesting that these lncRNAs could be used as prognosis markers. To date, current cancer diagnosis biomarkers are unreliable due to the high number of false positive and false negative results. Yet, since lncRNAs have highly specific expression patterns, their background in biological samples is virtually nonexistent and their presence can be detected in biological fluids [189,242,243,244,245,246]: whole blood [247], urine [248,249], serum [250,251], saliva [252], and gastric juice samples [253].

Their high specificity allows for the detection of cancer initiation (e.g., *SPRY4-IT1*) [254], progression (e.g., *ATB*) [255], metastasis (e.g., *LINC00461*, *CCAT2* and *H19*) [256,257,258] and response to therapy (e.g., *MALAT1* or *HOTAIR*) [77,259,260]. An additional important group of lncRNAs is the exosomal lncRNAs—secreted by cancer cells. They might play pivotal roles in tumorigenesis (e.g., *H19* in cervical cancer and hepatocellular carcinoma, *ZFAS1* in gastric carcinoma) and might be excellent cancer biomarkers (e.g., *MALAT1*, *HOTAIR*, both in cervical and bladder cancer). The functions, if any, of these exosomal lncRNAs are yet to be unravelled [261,262,263].

As lncRNA can be easily amplified and quantitated, this equals high sensitivity and specificity, easy and minimally invasive sample collection in contrast with conventional biopsies, and inexpensive methods [264]. A future perspective is to make them suitable for routine procedures in clinical practice and use for diagnostic purposes.

The therapeutic potential of lncRNAs is another crucial aspect. Recent reports have highlighted the relationship between lncRNA dysregulation and resistance to chemotherapy and targeted therapy [265,266,267,268], inhibition of signal transduction [15,269], and resistance to anti-hormone therapies [265,268,270]. For instance, overexpression of *HOTAIR* enhances the proliferation of breast cancer cells, while its depletion by shRNA considerably reduces cell survival and decreases growth during anti-hormone therapy (tamoxifen) [74]. Similarly, Adriaens and collaborators [271] demonstrated that the sensitivity of precancerous cells to DNA damaging or chemotherapy drugs was enhanced by knockout lncRNA-*Neat1* in mice and *NEAT1* in MCF-7 cells. Furthermore, the depletion of potentially oncogenic lncRNA *SNHG15* by siRNA or CRISPR-Cas9, decreased cell proliferation and invasion, and tumorigenic capacity of CRC cells while overexpression of this lncRNA directed to opposite phenotype [272]. Similarly, deletion of *LINC00538* (*YIYA*) in human breast cancer strongly inhibited the tumour growth and invasion in vitro and suppressed tumour growth in a mouse xenograft model [273].

Therefore, many potential oncogenic lncRNAs are currently being tested as targets for therapy (Figure 6). For example, targeting lncRNA *MALAT1* with ASO induced differentiation and inhibited metastasis breast cancer in the mouse mammary tumour virus-PyMT (MMTV) carcinoma model [274,275]. The anti-metastatic effect of knocking down *MALAT1* was also reported in a lung cancer xenograft model [276]. Silencing of a lncRNA called Cancer Susceptibility Candidate 9 (*CASC9*), who is associated with various processes in several cancer types, provided conflicting results, as it decreased the migration and invasion potential of ESCC (Esophageal Squamous Cell Carcinoma) cells while promoted their apoptotic potential of breast cancer cells MDA-MB-415 and MCF-7/DOX [277].

The comprehensive studies on lncRNAs associated with Notch oncogenic signalling in T-cell acute lymphoblastic leukaemia (T-ALL) uncovered a novel target candidate for treatment of this aggressive haematological disease, namely *LUNAR1* (leukaemia-induced non-coding activator RNA 1) [115]. Silencing *LUNAR1* in T-ALL cells suppressed the expression of IGF1R gene crucial for T-ALL tumour growth and caused significant cell growth retardation. Recently, *LUNAR1* was found expressed in colorectal cancer (CRC), expanding the promise of *LUNAR1* as a therapeutic target [278].

Targeting lncRNAs might not be straightforward; nevertheless, the removal of regulatory lncRNAs might cause complex changes in the cell. This was illustrated when the Colon Cancer-Associated Transcript 1 (*CCAT1*) was erased by CRISPR/Cas9, which led to the dysregulation of genes involved in several biological processes, including metabolism, cell migration, proliferation; and *CCAT1*-null cells lost their ability to anchorage-independent growth [279].

In addition to silencing a lncRNA, also adding or replacing it might have clinical benefits. In chronic metabolic disease such as atherosclerosis, lncRNA *LeXis* (Liver-expressed LXR-induced sequence) gene therapy resulted in positive outcomes. Studies on mice showed significantly reduced atherosclerotic load in mice treated with a vector expressing *LeXis* compared to control (the vector expressing Green Fluorescent Protein) treated mice [182]. Taking into account that there is an orthologue of *LeXis* in humans, such data might have clinical implications.

### 9.2. Silencing of Single Chromosome/Trisomy Effects

As described above, XCI is the result of a lncRNA—*XIST* activity. With this in mind, Yiang and collaborators developed a system to silence a single chromosome in a trisomy with the use of *XIST* [280]. They introduced an inducible XIST transgene to one of the three Chr21 in Down Syndrome (DS) patient-derived pluripotent stem cells using zinc finger nucleases. Induction of *XIST* expression resulted in the repression of virtually all genes across the autosome, and a total chromosome 21 transcriptional output near normal disomic levels which also rescued the cells phenotype [281]. That was the pioneering work on „chromosome therapy”, a technique that could have implications in many other chromosomal disorders, some of which are fatal in the first 1–2 years of life (e.g., trisomy Chr13 and Chr18).

### 9.3. Tissue/Muscle Regeneration

In a series of unexpected discoveries, lncRNAs have been found to modulate stem/progenitor cells physiology, including cells for engineering tissues. Growing evidence supports the importance of lncRNAs in different cellular lineages (neuronal, liver, skin, muscle and vascular tissue) growth, maintenance, proliferation, migration, and differentiation (for review see [282]), which suggests that lncRNAs can be applicable for tissue engineering. For example, lncRNA *Dum* (Developmental pluripotency-associated 2 (Dppa2) Upstream binding Muscle lncRNA) is induced during the early regeneration stage, when satellite cells become activated, proliferate and start to differentiate [283]. *Dum* acts in cis by silencing transcription of its upstream neighbour gene, Dppa2, encoding a pluripotency regulator, promoting then early differentiation. *Dum* interacts with and recruit DNA methyltransferases (Dnmts) to Dppa2 promoter, leading to CpG site hypermethylation and Dppa2 gene silencing. Dum knockdown in vivo impaired the injury-induced muscle regeneration, suggesting an important role of this lncRNA in the regulation of myogenesis [283]. Therefore, the careful and coordinated regulation of expression of lncRNAs, might be essential to engineer more physiologically relevant tissues for transplantation.

In terms of the bigger picture, extensive research on lncRNAs structure, partnerships and function might provide powerful outcomes in the future for different branches of medicine supporting therapeutic modalities for various pathogen infections, anticancer therapy, “chromosome therapy” and transplantology.

## 10. Discussion

Since their discovery, we have made great progress in our understanding of lncRNAs, as numerous studies have documented the significant contribution of lncRNAs to multiple regulatory mechanisms essential for mammalian homeostasis and development. However, the detailed functions of lncRNAs remain largely unknown. ‘Decoding’ lncRNAs in terms of revealing the key aspects determining their function is an ambitious goal due to their low expression in the cell, flexible structure, weak sequence conservation and diversity of functions.

The structural architecture of lncRNAs is still an unexplored field. With the rise of sequencing techniques such as single-cell RNA-seq (scRNA-seq), many cell-type-specific lncRNAs have been identified, and it is likely that many structural architectures and mechanisms will be revealed. Initially, the identification of common short sequence elements, similar structural features and structure-function relationships in lncRNAs might be applied. The resolving of lncRNA structure will need a combination of high-resolution global imaging techniques with secondary structural determinations. Moreover, the plasticity of all RNA molecules dynamically responding to the presence of their interacting partners, cellular context and environmental conditions are factors that cannot be ommited. While it is notoriously difficult to crystallize RNA molecules, cryoEM seems to work very well for RNA, and it negates the need for crystals, requires minimal sample and can differentiate biochemical and conformational heterogeneities of the samples. Thus, applying this powerful technique might enable the breakthrough in lncRNA studies similarly to the revolutionizing the structural biology of macromolecular protein assemblies, in the future. Structural maps can then be used to guide detailed biophysical explorations of lncRNAs cell functions and study the effect of mutations on lncRNA activity as well as elucidating their involvement in disease occurrence and designing lncRNA- targeting drugs.

Though the number of platforms for annotating lncRNA functions is growing, the highly efficient methods with enhanced targeting, higher resolution and increased maneuverability at the molecular level still need to be developed. The assessment of lncRNAs functions in cells requires a thorough analysis including gene perturbation experiments such as overexpression and downregulation, followed by real-time quantitative PCRs or deep sequencing, to observe any differential gene expressions. Importantly, not all lncRNAs exhibit functions in cells and laboratory model animals or are difficult to knock down or overexpress due to their enormous sizes and/or genomic architecture.

Understanding the structural biology and mechanism of action of lncRNAs will give the necessary foundation of the identification of therapeutics targeting these molecules with high specificity. Presently, the most popular method to target lncRNAs is through oligonucleotide-based therapies, yet CRISPR-Cas9-based techniques are also of a great potential. Importantly, the interactive network between lncRNAs and their partners, including other noncoding RNAs, particularly miRNAs, is another research field that should be explored in future studies. This complex relationship might also alter the disease’s level occurrence, course, or response to therapy. As mechanisms of action, the design of oligonucleotides and methods for targeting lncRNAs continue to be researched it is expected that a number of compounds efficient for lncRNA-based treatment of a wide variety of diseases will be obtained.

## 11. Conclusions

LncRNAs represent a new class of RNA that fine-tune complex physiological processes and the onset of diseases. The detailed understanding of their functions, structures and interactomes is challenging as the conventional methods used to study mRNA functions are inefficient for lncRNAs. In this review we summarized the interdisciplinary efforts undertaken to characterise lncRNAs. In many cases, a combination of several techniques was successfully applied; for example, for predicting structure of these molecules the bioinformatic predictions, followed by enzymatic or chemical probing of lncRNAs and HTS. Moreover, the ability of lncRNAs to interact with DNA, RNA, and proteins to exert their functions, is raising their complexity to a higher level. To explore the interactome of lncRNA techniques based on crosslinking RNAs with their partners, employing known RBPs, labeling RNA–protein complexes or adapting chromatin IP with oligonucleotide probing of RNA are coupled with sequencing or MS, depending on the target molecules. Yet, the nature and dynamics of such interactions need to be elucidated in the future. Functional studies on lncRNAs required new tools dedicated for their versatile activities in the cell. For the determination of lncRNAs function and map functional elements currently used methodologies target certain regions or whole lncRNAs gene through RNAi, ASOs, PNAs or CRISPR/Cas9. The improvement of these methods is of great value, as lncRNAs are connected with a wide spectrum of diseases. Exploring their biology, disease-related lncRNAs will gain greater relevance as potential biomarkers in cancers and for personalized medicine, especially for gene therapy.

## Figures and Tables

**Figure 1 cancers-13-02643-f001:**
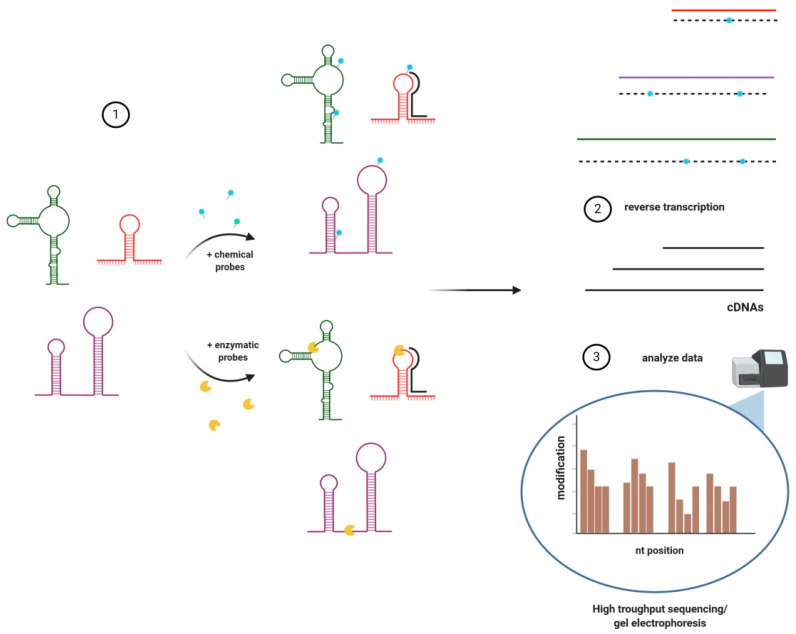
Chemical/enzymatic methods for lnRNAs structure analysis. (**1**) Probing lnRNAs by chemical probes which form covalent adducts, or by enzymes; (**2**) reverse transcription of modified or fragmented lnRNAs; (a) ‘stop’ event (blue dot) occurs when RT (reverse transcriptase) stalls or dissociates before adduct, resulting in a truncated fragment; (b) ‘mutation’ event occurs when RT proceeds through the adduct introducing a mutation; (**3**) detection using mutation or fractionation readout approaches. This is an original figure created in BioRender.com (accessed on 25 May 2020).

**Figure 2 cancers-13-02643-f002:**
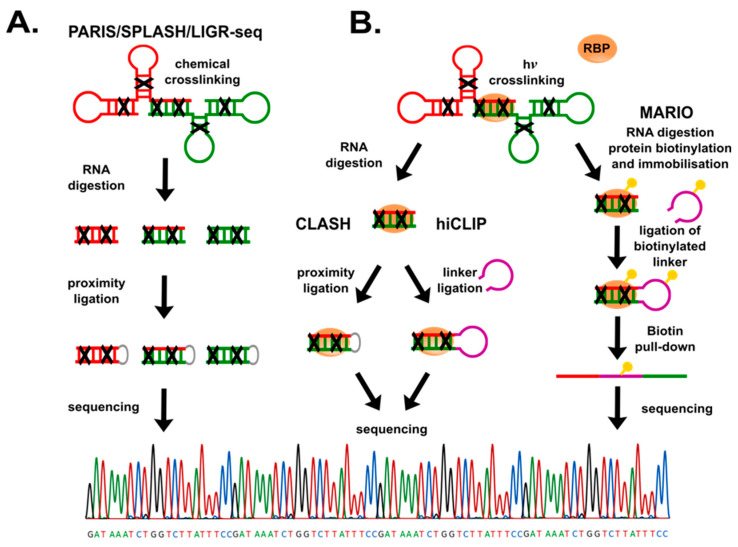
Methods for detection of RNA-RNA interactions. (**A**) Crosslinking RNAs directly, (**B**) crosslinking RNAs in the presence of RNA binding protein (RBP). PARIS, SPLASH, LIGR-seq include crosslinking RNAs with psoralen, digestion with RNase, proximity ligation and sequencing. In hiCLIP after RNA digestion, adjacent RNAs ends are ligated using a linker. In MARIO, RNA binding protein (RBP) is biotinylated, enabling immobilisation of RNA–RBP followed by digestion of RNAs ends and their ligation with added biotinylated linker; the tagged RNA–protein complexes are then purified and sequenced. This is an original figure.

**Figure 3 cancers-13-02643-f003:**
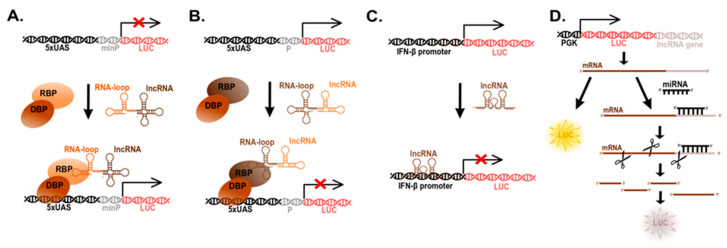
Reporter systems used to study lncRNA functioning as (**A**) transcriptional activator, (**B**) complex transcriptional repressor, (**C**) direct transcriptional repressor, (**D**) miRNA sponges. Abbreviations: UAS: upstream activation sequence, min P: minimal promoter, P: promoter, RBP: RNA binding protein, DBP: DNA binding protein, LUC: luciferase, IFN-β: interferon beta, PGK: phosphoglycerate kinase, promoter. This is an original figure.

**Figure 4 cancers-13-02643-f004:**
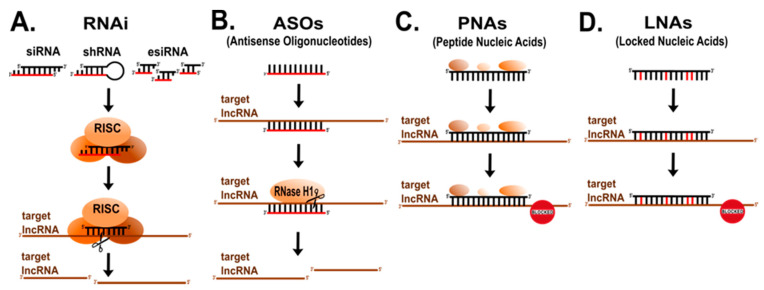
Targeting lncRNAs by RNAi, ASOs, PNAs, LNAs to knockdown and/or map lncRNAs functional domains. (**A**) RNA interference (RNAi) recruits the multiprotein RNAi-induced silencing complex (RISC) containing a siRNA (small interfering RNA), shRNA (short hairpin RNA) or esiRNA (endoribonuclease-prepared RNA) to specifically degrade the targeted RNA. (**B**) Antisense oligonucleotides (ASOs) bind to their targeted RNA, triggering endogenous RNase H1 to cleave the RNA/DNA heteroduplex. (**C**) Peptide Nucleic Acids (PNAs) and (**D**) Locked Nucleic Acids (LNAs), hybridize with complementary RNA sequences and affect activity of targeted lncRNAs. This is an original figure.

**Figure 5 cancers-13-02643-f005:**
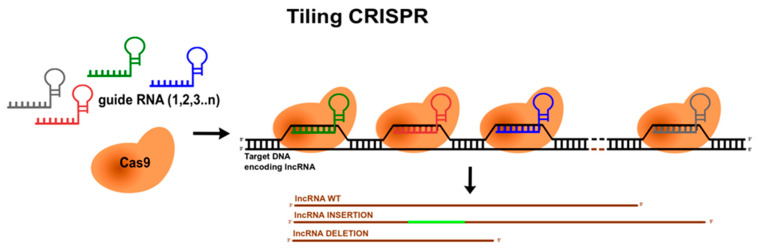
Schematic representation of Tiling CRISPR. Several guide RNAs target Cas9 endonuclease to sequences mapping wild-type (WT) lncRNA. As a consequence, insertions/deletions are introduced to lncRNA. This is an original figure.

**Figure 6 cancers-13-02643-f006:**
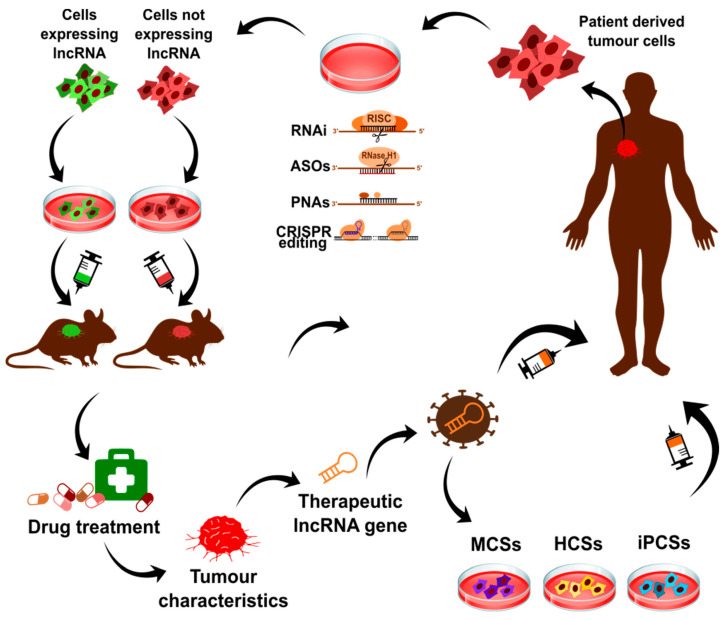
Schematic of anti-tumour gene therapy approaches involving lncRNA. The altered expression of lncRNA is widely observed in tumour tissues. The initial step for gene therapy includes collecting and growing tumour cells that are obtained from the patient (or from commercial cell line collections). The next step is testing the function of lncRNA of interest in tumour, which might be done with cell-editing assays using targeting lncRNA by RNAi, ASOs, PNAs or CRISPR/Cas9. Resulted cell lines with altered expression of lncRNA are characterized in line with unmodified tumour cells. The cell-based experiments include analysis of cell viability, proliferation, migratory potential, response to therapeutic compounds. The obtained results need to be confirmed in vivo for example after injection of lncRNA expressing cells/lncRNA-silenced cells to immunodeficient mouse (xenograft models). The tumour growing in xenografts mimics the patient’s tumour. The characteristics of tumour mass, growth rate and specific aspects of its behavior, such as assessing metastatic growth, is needed to validate the therapeutic potential of lncRNA. Additionally, xenografts might be subjected to anti-tumour therapy to test response to drugs. The verified therapeutic lncRNA gene might be then encapsuled with the non-immunogenic vectors like viruses and injected to the patient. Ultimately, stem cell lines such as MCSs (mesenchymal stem cells), HCSs (haematopoietic stem cells) or iPCSs (induced pluripotent stem cells) might be transfected with the lncRNA to obtain cells expressing lncRNA. The injection of these modified cells to the patient increases the ability to generate healthy cells. Several lncRNAs tested for gene therapy are described within the text. This is an original figure.

**Table 1 cancers-13-02643-t001:** Comparison of commonly used methods for lncRNA structural studies.

Type of Experiment	Limitations		Methods	Structure	lncRNA	References
In vitro	Controlled conditions, lncRNA structure more stable than in in vivo tests Requires sequencing		**Enzymatic probing**			
		Only in vitro Fragmentation of RNA by nucleases makes difficult the identification of secondary structures at single-nucleotide level Might require additional probing with nuclease or chemical	PARS/nuclease S1 and RNaseV1	double- and single- stranded regions of RNA	*Rox1* *Rox2*	[73]
			**Chemical probing**			
		Does not provide information concerning interactions between base-pairing (at close- or long-range) Labour-intensive	SHAPE-seq	2′-OH acylation	*Braveheart RepA* *Rox1* *Rox2* *SRA HOTAIR COOLAIR MALAT1 NEAT1*	[66,67,68,69,71,73,74,75]
		DMS- nucleotide bias (only able to react with purines) DMS is corrosive and toxic, In some cases, using DMS is not sufficient to capture all single-stranded regions Labour-intensive	DMS-seq (DMS)	unpaired adenine and cytosine residues	*BraveheartRepA* *SRA HOTAIR MALAT1*	[66,67,69,74,75]
In vivo	Requires sequencing Difficult task due to lncRNAs’ size and low abundance in cells lncRNAs are expressed in alternative isoforms and bound by a variety of RNA binding proteins in vivo		**Chemical Probing:**			
			SHAPE-MaP (1M7,1M6,NMI1)	2′-OH acylation	*Xist*	[55]
In silico	lncRNA is more structured than in in vivo tests Only for prediction of secondary structure Has to be complemented by in vivo/in vitro tests		CROSS (Computational Recognition of Secondary Structure)	RepA, D2 domain	*Xist HOTAIR*	[76]
Biophysical	Difficult for application of higher numbers of transcripts Difficult for long RNA strands Requires large quantities of robust, homogenous sample Crystallography is labour-intensive		X-ray	A-rich 3′-UTR	*MALAT1*	[77]
			NMR spectroscopy	AUCG tetraloop	*Xist*	[56]

**Table 2 cancers-13-02643-t002:** Commonly used RNA binding proteins (RBPs). Abbreviations: H = A, C, or U; N = A or U; Y = U or C.

RBP	RNA Sequence/ Motif	Interacting RNA	Method	References
λN	Box B loop	any fused with Box B loop	Gal4-λN/BoxB reporter system	[90,91]
MCP	MS2 loop	any fused with MS2 loop	RNA-tethering	[92]
IGF2BP1,2,3	CAUH	mostly exons, i.e., eEF2	PAR-CLIP	[84]
PUM2	UGUANAUA	3′ untranslated region (UTR)	PAR-CLIP	[84]
QKI	ACUAAY	mostly introns	PAR-CLIP	[84]
AGO (most enriched 7 nucleotide- mers)	AUGCUGC	miR-103,-107	PAR-CLIP	[84]
	GCUGCUA	miR-15a/b,-16,196a	PAR-CLIP	[84]
	UUUGCAC	miR-19a/b	PAR-CLIP	[84]
	UGCACUU	miR-130a/b,-148a/b,-301a/b	PAR-CLIP	[84]
	CACUUUA	miR-106a/b,-20a/b	PAR-CLIP	[84]
	UUGCUGC	miR-424	PAR-CLIP	[84]
	UUGCACU	miR-130a/b,181a,-301a/b,-454	PAR-CLIP	[84]
	GCACUUU	miR-17,-20a/b,-93,-106a/b	PAR-CLIP	[84]
	UGCUGCU	miR-15a/b,-16,196a,-103,107,-424	PAR-CLIP	[84]
STAU1	3′ UTRs (*Alu*, 858 nt duplex)	*XBP1*	hiCLIP	[87]

**Table 3 cancers-13-02643-t003:** Examples of tissue specific lncRNAs.

Tissue	lncRNA	References
Brain (frontal cortex’s white matter)	*OLMALINC* *TUNA*	[176] [177]
Heart	*Braveheart*	[178]
Lung	*HSPC324* *MALAT1*	[179] [180]
Pancreas	*MALAT1*	[180]
Liver	*lncLSTR* *LeXis*	[181] [182]
Testis	*THOR*	[183]
Muscle	*Linc-MD1*	[184]
Skin	*ANCR* *TINCR*	[185] [186]

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
