# Peer review of "Decoding LncRNAs"

_cancers, 2021, doi:10.3390/cancers13112643_

Round 1
Reviewer 1 Report
This is a review article summarizing updated knowledge on decoding lncRNAs. The article provides critical analysis on the achievements of lncRNA studies with very nice illustration on lncRNA interactomes, structural studies, and methodology of reporter approaches for lncRNA functional elements. The article also provides the author’s view on the clinical applications of targeting lncRNAs for treatment of human diseases. The manuscript is well prepared. The generated knowledge should be valuable for the researchers to push the field forward. I have only a few minor comments to help the authors to improve the quality of the manuscript.
- What has known about diversity of lncRNAs in lengths and how different lengths can affect lncRNA structure and their functions?
- At the end of the manuscript, authors should write a paragraph to discuss major problems remaining the area of lncRNAs and how the knowledge gaps hinder further development in the field.
- A better discussion on future directions should be explored to address the identified knowledge gaps and current challenges.
- A paragraph for main conclusions on the reviewed topics and key concepts may help readers to get a clear picture on current status of the field.
- The legend of Figure 6 should be expended to provide details on each step for anti-tumour gene therapy approaches.
- In figures, first letters of labels should be capitalized.
Author Response
Response to Reviewer 1 Comments
We would like to thank the Reviewer for thorough revision of the manuscript and constructive remarks. We believe that implementing the Reviewers suggestions into the publication will significantly improve and clarify the manuscript. Please find below a detailed point by point response to all comments.
Point 1: What has known about diversity of lncRNAs in lengths and how different lengths can affect lncRNA structure and their functions?
Response 1: We are very grateful for pointing that issue. Accoding to the Reviewer’s suggestion we have added information about diversity of lncRNAs in lenths and its relation to srtucture/function of these molecules in the section 2. LncRNAs structure-functions page 3.
Point 2:
At the end of the manuscript, authors should write a paragraph to discuss major problems remaining the area of lncRNAs and how the knowledge gaps hinder further development in the field.
Point 3:
A better discussion on future directions should be explored to address the identified knowledge gaps and current challenges.
Point 4:
A paragraph for main conclusions on the reviewed topics and key concepts may help readers to get a clear picture on current status of the field.
Responses 2-4:
We agree with this suggestions. We have extended the manuscript adding at the end of it two sections: 10. Discussion (page 20) and 11. Conclusions (pages 20-21).
Point 5:
The legend of Figure 6 should be expended to provide details on each step for anti-tumour gene therapy approaches.
Response 5:
We have corrected the legend for the Figure 6 according to this suggestion (page 18).
Point 6:
In figures, first letters of labels should be capitalized.
Response 6:
To prepare the manuscript we have used the Microsoft Word template according to the instructions for submission in Cancers, which recommend using format Figure X to mark figures. Therefore, we have not corrected the format in this version of the manuscript.

Reviewer 2 Report
The manuscript regarding the decoding of lncRNA sequence, structure and functions in humans, both normal or in disease, is really interesting. The idea of presenting all experimental methods that lead to deciphering lncRNA features is really original. However, there are aspects that can additionally further improve the manuscript.
Since the most of papers regarding lncRNA involvement in e.g. cancer development begins with in silico assessment of the possible lncRNA actions, it would be great to introduce additional chapter that describes the recently known and accessible free online tools and databases for lncRNA annotation and perdicted funtions. Also, the paper needs proper, more profound ending chapter with conclusions and future perspectives.
More issues are listed below:
- line 33: the reference for the 50 000 human lncRNA is needed;
- lines 126-126: 'While lncRNAs do not seem to be evolutionary conserved', then lines 130-131: 'the 3D structure of the resulting RNA, and their expression patterns, support that probably over 70% of 5413 human lncRNA analysed are evolutionary conserved' This part is contradictory. lncRNAs, although low sequence similarity, they are still highly conserved regarding the secondary sequence and function. Please rewrite this;
- throughout the manuscript, the authors use 'et al.', 'and coworkers' or ' and collaborators'. The manuscript should be consistent in this regard;
- figure 2 on line 379 and figure 2B on line 382 should be changed to figure 3 and 3B, respectively;
Author Response
Response to Reviewer 2 Comments
We are very grateful for critical revision of the manuscript, all detailed comments and remarks. Following the Reviewers suggestions, we have improved the manuscript. We hope that we have responded adequately to them below.
Point 1: Since the most of papers regarding lncRNA involvement in e.g. cancer development begins with in silico assessment of the possible lncRNA actions, it would be great to introduce additional chapter that describes the recently known and accessible free online tools and databases for lncRNA annotation and perdicted funtions. Also, the paper needs proper, more profound ending chapter with conclusions and future perspectives.
Response 1: We wan’t to thank the Reviewer for this suggestion. We have summarized the available on-line tools for analysis of lncRNAs in a new section 7. LncRNAs databases and bioinformatic tools (page 14-15). Also, we have extended the manuscript adding at the end of it two sections: 10. Discussion (page 20) and 11. Conclusions (pages 20-21).
Point 2:
line 33: the reference for the 50 000 human lncRNA is needed;
Response 2:
We have added the proper reference.
Point 3:
lines 126-126: 'While lncRNAs do not seem to be evolutionary conserved', then lines 130-131: 'the 3D structure of the resulting RNA, and their expression patterns, support that probably over 70% of 5413 human lncRNA analysed are evolutionary conserved' This part is contradictory. lncRNAs, although low sequence similarity, they are still highly conserved regarding the secondary sequence and function. Please rewrite this;
Response 3:
We agree that this issue should be clearly presented. We have edited the text according to this remark (page 4).
Point 4:
throughout the manuscript, the authors use 'et al.', 'and coworkers' or ' and collaborators'. The manuscript should be consistent in this regard;
Response 4:
We have corrected the text to unify the in line citations (using “and collaborators”).
Point 5:
figure 2 on line 379 and figure 2B on line 382 should be changed to figure 3 and 3B, respectively;
Response 5:
We have corrected the text according to this suggestion (page 11).
